# Development of spontaneous vegetation on reclaimed land in Singapore measured by NDVI

**Leon Yan-Feng Gaw**[1,2]⊛*, **Daniel Rex Richards**[1,2]⊛

**1** Natural Capital Singapore, Singapore-ETH Centre, ETH Zurich, Singapore, Singapore, **2** Campus for Research Excellence and Technological Enterprise, Singapore, Singapore

⊛ These authors contributed equally to this work.
* gaw@arch.ethz.ch

**Data Availability Statement:** The data underlying this study are available on Figshare (https://figshare.com/articles/dataset/Data_from_Development_of_spontaneous_vegetation_on_

## Abstract

Population and economic growth in Asia has led to increased urbanisation. Urbanisation has many detrimental impacts on ecosystems, especially when expansion is unplanned. Singapore is a city-state that has grown rapidly since independence, both in population and land area. However, Singapore aims to develop as a 'City in Nature', and urban greenery is integral to the landscape. While clearing some areas of forest for urban sprawl, Singapore has also reclaimed land from the sea to expand its coastline. Reclaimed land is usually designated for future urban development, but must first be left for many years to stabilise. During the period of stabilisation, pioneer plant species establish, growing into novel forest communities. The rate of this spontaneous vegetation development has not been quantified. This study tracks the temporal trends of normalized difference vegetation index (NDVI), as a proxy of vegetation maturity, on reclaimed land sensed using LANDSAT images. Google Earth Engine was used to mosaic cloud-free annual LANDSAT images of Singapore from 1988 to 2015. Singapore's median NDVI increased by 0.15 from 0.47 to 0.62 over the study period, while its land area grew by 71 km$^2$. Five reclaimed sites with spontaneous vegetation development showed variable vegetation covers, ranging from 6% to 43% vegetated cover in 2015. On average, spontaneous vegetation takes 16.9 years to develop to a maturity of 0.7 NDVI, but this development is not linear and follows a quadratic trajectory. Patches of spontaneous vegetation on isolated reclaimed lands are unlikely to remain forever since they are in areas slated for future development. In the years that these patches exist, they have potential to increase urban greenery, support biodiversity, and provide a host of ecosystem services. With this knowledge on spontaneous vegetation development trajectories, urban planners can harness the resource when planning future developments.

## Introduction

Urban growth is one of the leading drivers of ecological destruction in the world today [1]. As the global population increases, large numbers of people have migrated to cities in search for work and better economic opportunities [2]. Urbanisation has thus led to a simultaneous

reclaimed_land_in_Singapore_measured_by_
NDVI_/13550063).

**Funding:** This research is supported by the
National Research Foundation, Prime Minister's
Office, Singapore under its Campus for Research
Excellence and Technological Enterprise (CREATE)
Programme (NRF2016-ITC001-013).

**Competing interests:** The authors have declared
that no competing interests exist.

densification and expansion of urban coverage, to accommodate a growing population. The
expansion of built cover has come at a cost to the natural environment, with surrounding eco-
systems removed to provide space and building materials for growth [3]. Many urban centres
are located along coastlines for transport access and trade [4]. Coastal cities often expand land-
wards into natural terrestrial ecosystems, and also seawards into marine ecosystems [5]. Land
reclamation from the sea is an expensive solution to create land for urban growth. Thus, it is
mainly used in highly space constrained locations with high land prices, or to support essential
infrastructure such as sea- or airports [6]. Urban coastal reclamation has been pursued in
some densely-populated coastal cities in Asia, notably in Hong Kong, Macau, and Singapore
[5]. While land reclamation has a clear economic driver and is usually intended for urban
growth, the timeline of construction on newly reclaimed land can stretch over several decades
[7]. The slow pace of urban development on reclaimed land is partially due to the need to let
the soil settle and become firm before constructing skyscrapers. Moreover, it is only economi-
cally viable to reclaim large areas at once, so the supply of reclaimed land may temporarily out-
strip demand after the completion of a reclamation project [5]. Hence, it is common for
reclaimed land to be landscaped as public park space, or left fallow, for many years before
building construction begins [8].

Reclamation is intended to provide space for urban development in the long term, but may
also temporarily add valuable vegetated cover to urban centres. Urban vegetation brings a
wealth of benefits, or 'ecosystem services', to cities, such as cooling the air [9], regulating flood
risk [10], and providing habitats for biodiversity [11]. There is increasing interest in under-
standing how timelines of urban development and redevelopment can be best optimised to
provide for a shifting mosaic of urban vegetation, to maintain the provision of their benefits to
urban residents [12]. Reclaimed land that develops vegetation, either through natural colonisa-
tion or planting, could thus be an important source of vegetated cover, particularly in highly
land-scarce urban centres. Planted urban vegetation develops along a predictable trajectory,
but has considerable setup and maintenance costs attached to it [13], and may perform less
well in providing ecosystem services compared to natural vegetation [14]. On the other hand,
naturally established vegetation could provide a cost-effective way of providing high-value veg-
etation on reclaimed land, even though the development of vegetation in new habitats can be
highly unpredictable [15]. To effectively use spontaneously-generated vegetation on reclaimed
land when planning urban developments, a better understanding of the rates at which it estab-
lishes and develops is required.

Land use in urban areas has been monitored continuously since the launch of the first earth
observation satellites in space [16]. Initially the spatial resolutions of images were at 1 km; low
by today's earth observation standards and not useful for analysing the highly heterogeneous
landscapes found in urban areas [16]. The launch of Landsat by NASA in 1970s marked the
start of more constant and high resolution earth monitoring [17]. The Landsat sensors have
been instrumental in observing geographical phenomena including natural hazards, and
changes to earth's surface due to human development [18]. Due to the high volumes of data
captured and archived regularly and over a long period, land use changes can now be mea-
sured not just at two snapshots of time, but continuously across many time intervals [19].
Time series analysis of land and vegetation cover changes allow researchers to understand the
overall trends and dynamics of land use changes across complete time series instead of a sim-
ple increase or decrease between two points in time. Modelling of land use changes, for exam-
ple, vegetation growth rates, can be done more accurately using a dense time series of repeated
satellite images [20].

Historically, the large volumes of data provided by continuous satellite monitoring pro-
vided a technical challenge for interpretation and analysis. Processing of large time series

datasets has been made accessible through recent technological advancements in parallel computing, notably the creation of Google Earth Engine [21, 22]. In Google Earth Engine, data storage and processing are done on a cloud computing platform, giving access to unprecedented computing power to individual researchers [22]. The availability of Landsat imagery in Google Earth Engine's Data Catalog stretching back to 1985 now makes it possible to analyse changes to the earth's surface such as land reclamation and associated vegetation development.

This study aims to investigate the growth of pioneer species of vegetation on newly reclaimed land from the sea in Singapore; a nation that has pursued an ambitious land reclamation strategy that increased its land area by a quarter since 1963 [7]. Specifically, the objectives of this study were to:

1. Identify where in Singapore vegetation has grown on reclaimed land. This was achieved by compositing annual cloud free maps of Singapore's land area and NDVI.

2. Quantify the rate of spontaneous vegetation development for five newly reclaimed sites, using NDVI as a proxy. This was achieved by modelling temporal trends in NDVI across years.

## Materials and methods

### Study site

Singapore is a city-state in Southeast Asia located at the southernmost point of Peninsula Malaysia. As of 2018, Singapore has a land area of 742.22 km$^2$ that supported a population of 5 million [23]. The country is highly urbanized and has a population density of 7,804 people per km$^2$ [24] and is the third most densely populated country in the world [25]. The city has experienced rapid urban growth and urbanisation in the 20th century with colonisation, entrepot trade, industrialisation, and now a move towards a service economy. Urban growth is expected to continue its trend into the 21st century, as outlined in a governmental white paper that details plans for the city to accommodate a population of 6 million by 2030 [26].

Singapore is also an island, which means its territorial boundaries are constrained by the sea—the Straits of Johor in the north, and the Straits of Singapore in the south. Geospatial data from OpenStreetMaps [27] show that Singapore has approximately 1,400 km$^2$ of space that includes both land and sea areas. Previous work has estimated Singapore's land area in 2018 to be approximately 750 km$^2$ [23], leaving a sea area of 650 km$^2$. The sea space of Singapore is congested, comprising shipping lanes for cargo vessels, space for vessels to moor while waiting to dock at Singapore's ports, space reserved for recreation and sea sports [28], and a marine nature park [29]. Hence, not only is land area in Singapore important for national development, but the sea area is also critical to the economy for Singapore to be competitive in the shipping industry.

Singapore has constantly pursued land reclamation works since independence in 1965 to expand its land area for development. The process of reclamation began with infilling of mangrove forests using material taken from the tops of surrounding hills [28]. Later, Singapore began importing sand from nearby countries including Malaysia, Indonesia, and Cambodia [30]. The land area has thus expanded by 87 km$^2$ in 70 years [31] mainly in the south and east of Singapore. According to the latest Concept Plan 2011 published by the Ministry of National Development (MND) [32], Singapore plans to grow its land area in the southwest in Tuas, the east in East Coast and Changi, and the north-eastern islands of Pulau Ubin and Pulau Tekong. Reclaimed land in Singapore requires time to settle before urban development begins. In the case of one of the recently built-up reclamation regions—the Marina Bay area—it took 40 years for skyscrapers to emerge from space that once was the sea [33].

**Table 1. Summary of new land reclaimed from the sea and areas with high vegetation cover.**

| Data | Source | Sensors | Years |
|---|---|---|---|
| NDVI | Lacerda (2019) | Landsat 5, 7, 8 | 1988, 1990–1991, 1995–1997, 2000–2015 |
| Land Area | Pekel et al. (2016) | Landsat 5, 7, 8 | |

## Data acquisition

Two geospatial datasets were needed to annually map the (1) land area of Singapore, and (2) vegetation cover of Singapore. First, annual land area in Singapore was derived from an existing dataset of the world's water surface area from 1988 to 2015 [34]. The dataset was created using 3,865,618 historical Landsat images, which were classified into areas of 'No Data', 'Not water', 'Seasonal water', and 'Permanent water' [34]. In our study, annual water cover for all years available were downloaded from Google Earth Engine and visually inspected in ArcGIS Desktop 10.5 [35] to check for errors including artifacts due to insufficient data caused by cloud cover. The code used to extract land area and vegetation cover can be found in the online repository of the data hosted on Figshare. Cloud cover is a common problem in tropical regions such as Singapore [36] and this problem occurred more often with images taken before year 2000, due to the lower number of Landsat images available in these years [37]. After manually checking 27 annual water cover maps, 21 were deemed suitable for analysis (Table 1). Six years of land area images were excluded from this study because of insufficient cloud free images within the years of 1989, 1992, 1993, 1994, 1998, and 1999. The final year available in the dataset (2015) was used to identify areas of newly reclaimed land since 1988.

Second, to detect the presence of vegetation, the Normalised Difference Vegetation Index (NDVI) was downloaded for this study's years of analysis (Table 1). NDVI measures the greenness of vegetation by amount of chlorophyll detected in their leaves, and is considered a good indicator of vegetation health and status [38]. The higher the NDVI, the greener or more vegetated a place is. Annual composites of NDVI are applicable in tropical Singapore since there is neither winter-summer, nor major wet-dry seasonality [39], and vegetation is thus evergreen. In this study, NDVI was used as a measure of vegetation maturity in which the stages of vegetation growth are measured [40]. Code from Larceda [41] was used to generate cloud-free composite NDVI maps from 1988 to 2015. It starts with masking out cloud cover from Landsat 5, 7, and 8 images. Next, NDVI was calculated for the cloud-free Landsat images using their corresponding Red and Near-infrared spectral bands. Finally, all the cloud-free NDVI images from a year were composited together by taking the median of each pixel from that year to make a NDVI map of Singapore for that particular year. Years with no cloud free images of certain areas in Singapore resulted in patches of 'No Data' in earlier years of the NDVI mosaic. As with the land area images, six years of NDVI mosaics were excluded from this study (1989, 1992, 1993, 1994, 1998, and 1999).

## Data processing and analysis

Annual maps of water and land use were cross-referenced to identify new areas of reclamation between sequential pairs of years. Areas that were previously classified as 'Permanent water' and then became 'Not water' or 'Seasonal water' were reclassified as 'Newly reclaimed land' [42]. Areas of 'No Data' in [34] dataset were areas of land all throughout the 21 years of analysis. The classes that represented 'Newly reclaimed land' were then used as a mask to extract NDVI values within the reclaimed areas in each year [43]. Five sites of reclaimed land were selected for further analysis, as these sites are known to contain mainly spontaneous vegetation rather than intentionally planted park spaces [23]. They are Tuas, Jurong, Semakau, Changi, and Tekong. The relative years that each study site achieved 'Newly reclaimed land' status was

calculated by averaging the years that all the pixels within each study site changed from 'Permanent water' to 'Newly reclaimed land'. The mean and median years derived do not indicate exact dates land reclamation was completed but rather, the relative order of which land reclamation works were completed.

Pixels with NDVI values of equal or greater than 0.7 were extracted [44] to identify land with lush green vegetation established on it. The value of 0.7 was arbitrarily chosen, as there is no systematically agreed-upon value with which to measure full vegetation establishment [45, 46]. These pixels were each given a unique ID and were tracked for changes to NDVI values from 1988 to 2015, and the year when the pixel first changed to 'Newly reclaimed land' from 'Permanent water'. The results were then visualized in a line graph that shows the trend of NDVI values on newly reclaimed land and when the land was established. Temporal trends of NDVI on newly reclaimed land were modelled as linear mixed-effects models using the R statistical software [47] using the 'nlme' [48] and plotted with the 'sjPlot' [49] packages. NDVI development was modelled as a function of the year since the date of first reclamation, with the pixel identifier included as a random effect. A first-order temporal correlation structure was included in the linear mixed effects models, to account for potential temporal autocorrelation caused by repeated measurements at the same spatial locations [48]. In total, there were 16,968 pixels included in the analysis, translating to 15.27 km$^2$ of land area based on Landsat's spatial resolution of 30 m. Linear, Quadratic, and Logarithmic models were built and the most parsimonious model was selected by comparison using the Akaike Information Criterion (AIC) [50], R-squared statistics [51].

## Results

Singapore's land area expanded by 71.51 km$^2$ from land reclamation from the sea between 1988 to 2015. The areas of land reclaimed were mainly in the southwest, east, and the island of Pulau Tekong (Fig 1). The port area of Pasir Panjang and the oil refineries on Pulau Bukom also had land reclaimed, but no vegetation grew in these areas and they were therefore excluded from the NDVI component of this study. The entire area of Tuas in the southwest has been extended 14.05 km$^2$ southwards while the cluster of islands in the Jurong archipelago were combined to form one single Jurong Island, growing by 24.71 km$^2$ in size. The island of Pulau Semakau has been used as a landfill since 1995 [52], gradually infilled with incineration waste, thereby creating 2.45 km$^2$ in land area. The northwestern and southern areas of Pulau Tekong have grown in land area by 8.49 km$^2$. The Changi region experienced the largest increase in land area by 21.79 km$^2$.

Between 1988 to 2015, a total of 15.27 km$^2$ of high NDVI vegetation grew up on newly reclaimed land (Fig 2). By the end of the period, the reclaimed land in both Pulau Semakau and Pulau Tekong had more than 40% vegetated cover. Reclaimed land in the Jurong and Tuas regions had lower vegetated cover of 11.7% and 5.4%, respectively. In Singapore, the national median NDVI (measured by Landsat pixels located within Singapore's 1988 land area that excludes newly reclaimed land) has increased over 27 years at a rate of 0.0055 NDVI per year. The NDVI in reclaimed areas has increased at a more rapid rate than the national median (Fig 3). Apart from Jurong, all other sites showed a steep increase of NDVI after the first year, and gradually plateaued in NDVI value around year 15 (Fig 4). Modelling of the relationship between NDVI and year since reclamation showed a quadratic relationship (Table 2), with the NDVI value growing to 0.7 after 16.9 years.

## Discussion

### Vegetation and reclamation in Singapore

Reclamation has dramatically increased Singapore's land area over the past 30 years. Simultaneously, spontaneous vegetation growing on this reclaimed land has made a substantial

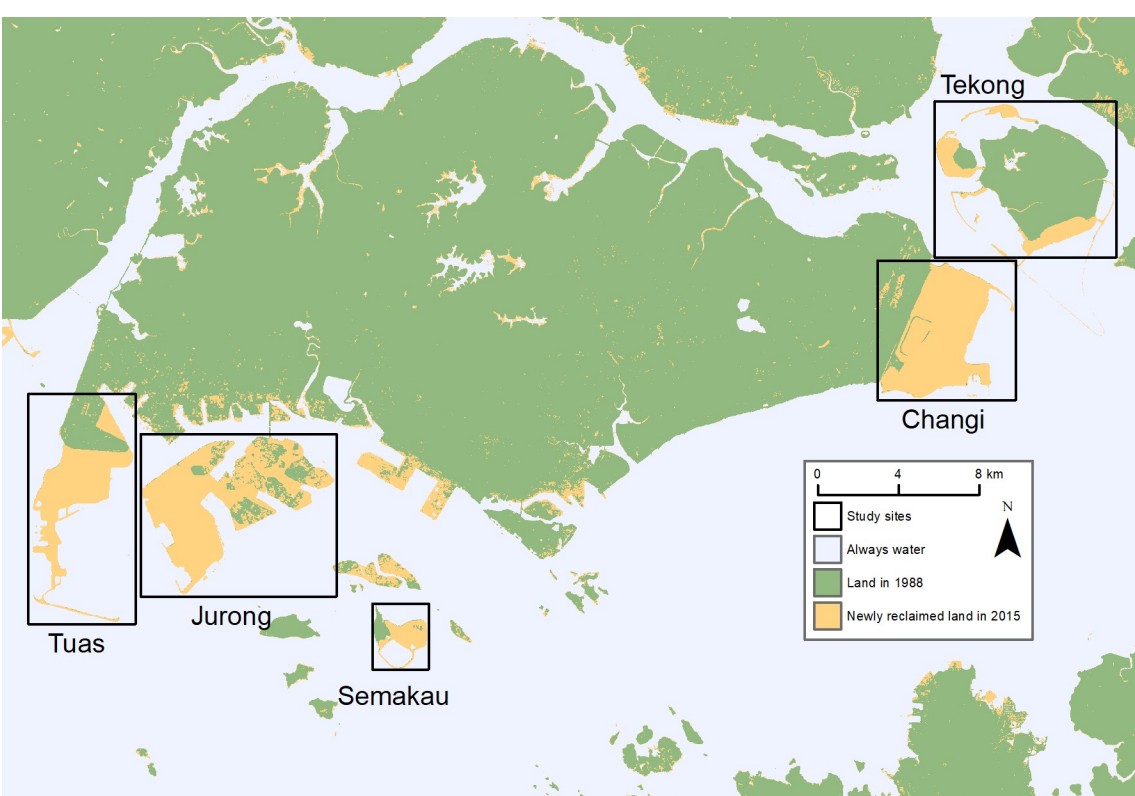

**Fig 1. Newly reclaimed land in Singapore between 1988 and 2015.**

contribution to increasing Singapore's vegetated cover, and median NDVI [53, 54]. The trajectory of development of urban vegetation is broadly similar across all sites (Figs 4 and 5), suggesting that the types of vegetation growing, and the environmental suitability for these plants are likely to be similar. This is not surprising, as Singapore's small area does not cover a wide range of environmental conditions [55], and the material used in reclamation is uniform across most of the historical reclamation works [53].

All reclaimed regions had a negative NDVI value immediately following reclamation (Fig 4), indicating bare land or land of a water logged or seasonal nature [34]. Despite the general trajectory of vegetation development observed across the sites, there was variation in urban green cover in 2015 (Fig 2). This variation is partly due to the contrasting dates of reclamation, but also the type of intended eventual land use and urgency of that use. The Tuas and Jurong sites showed lower green cover in 2015. In the case of Jurong, a group of thirteen islands [56] were combined to form the petro-chemical refining hub that occupies the entire island. Petrochemicals are important for Singapore's economy—therefore, facilities were built rapidly once land had been reclaimed, with little space given for natural vegetation to establish [57]. In addition, such industrial development typically requires low-rise rather than high-rise development, which has less stringent requirements on soil stability and structure, and so can proceed more rapidly after reclamation [58]. The Tuas site will be developed for port infrastructure in future, with much of the reclaimed area designated for temporary container storage, cranes and warehousing [59]. Such infrastructure typically involves little or no building above the ground, so can come into use rapidly following reclamation. A further explanation for the lack of vegetated cover at Tuas is that the date of reclamation at Tuas was relatively long ago,

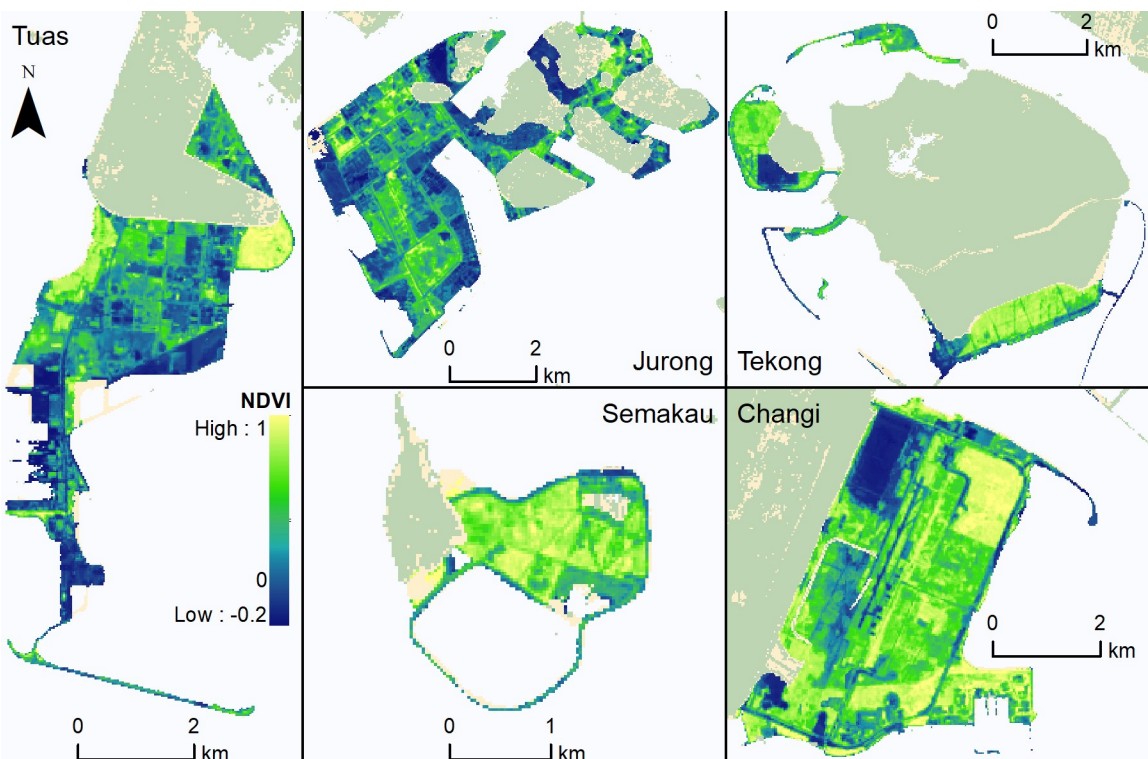

**Fig 2. NDVI values in 2015 on reclaimed land.** More yellow colours indicate higher NDVI values, with more blue colours indicating lower NDVI values.

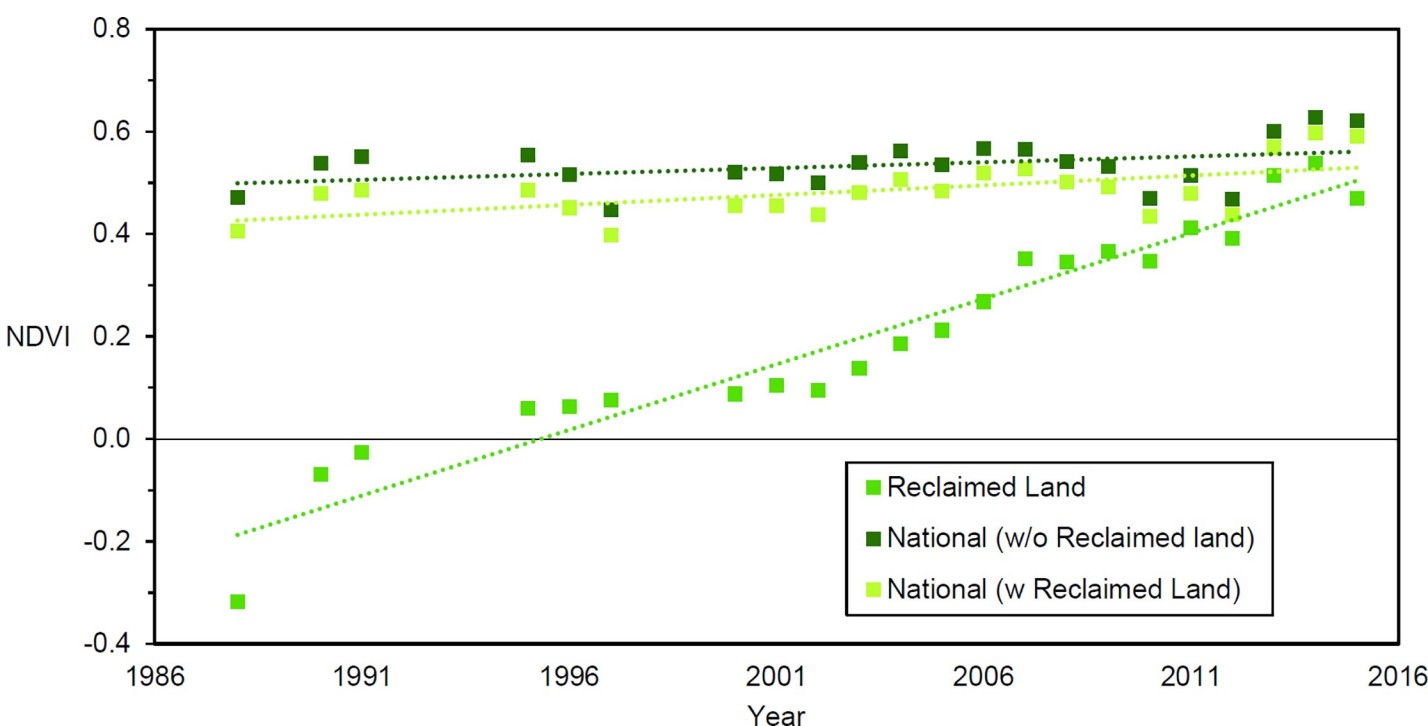

**Fig 3. The trend of NDVI across reclaimed areas only, Singapore without reclaimed areas, and Singapore with reclaimed areas across the study period.**

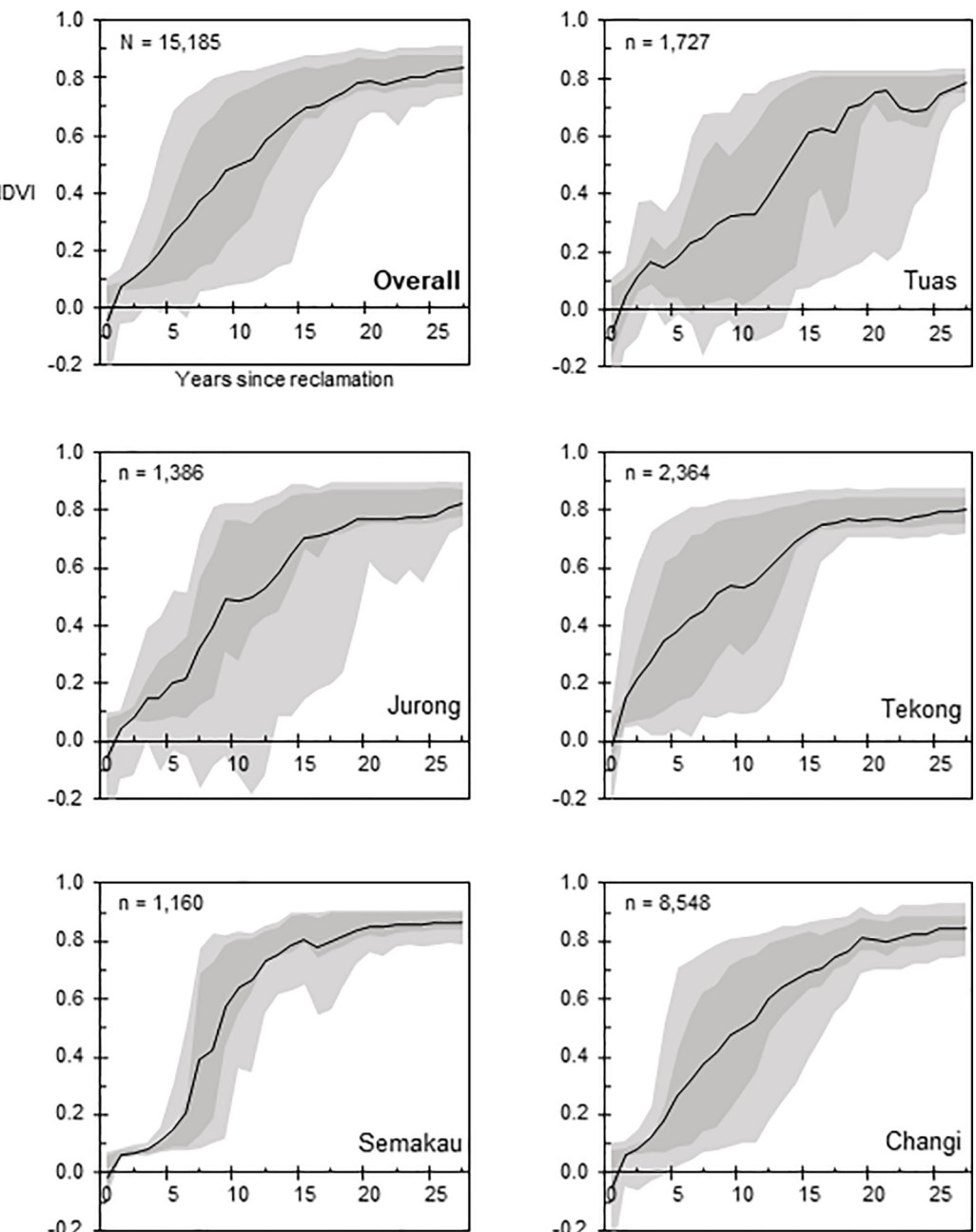

**Fig 4. Temporal trends of NDVI on reclaimed land, from the first year of reclamation from water.** Black lines indicate median. Dark grey shading indicates 50th percentile. Lighter grey shading indicates 80th percentile. 'N' in the graph denotes the total number of sample pixels while 'n' in the graphs denotes pixels at each study site.

leaving more time for infrastructure development to progress (Table 3). In some unvegetated areas of Tuas the land may not have been present long enough for any vegetation to establish.

At the other three sites, vegetation cover was relatively high. Two of these sites—Tekong and Semakau—are not located on the main island of Singapore, and are thus further from the main centres of urban development. These offshore reclamation sites are accessible only by shipping vessels and do not have any residential developments or permanent inhabitants [29].

**Table 2. The models evaluated for this study.** Model 2 was chosen due to its lower AIC and higher $R^2$ value.

| No | Y | X | Model | n* | AIC | Adj. $R^2$ | p |
|----|------|------|-------|-----|------|-----------|----|
| 1 | NDVI | Year | $Y = 0.0321 \cdot X + 0.0271 + \varepsilon$ | 425180 | -696087 | 0.59 | < 0.01 |
| 2 | NDVI | Year | $Y = 0.0698 \cdot X - 0.0031 \cdot X2 - 0.1069 + \varepsilon$ | 425180 | -723044 | 0.65 | < 0.01 |
| 3 | NDVI | Year | $Y = 0.2406 \cdot log(X) - 0.0282 + \varepsilon$ | 425180 | -704345 | 0.47 | < 0.01 |

*The 'n' here refers to the product of number of pixels with NDVI equal or greater than 0.7 (16,968) and the study period (27 years) with gaps in data removed.

Pulau Tekong is designated as a military training ground, while Pulau Semakau is designated as an incineration waste landfill site [60]. Access to these islands is restricted, which makes development on them less likely, especially when future master planning documents do not dictate any pressing need for development other than their current use [61, 62]. With less disturbance, vegetation has been left to grow unimpeded in these areas, with only limited disturbance from ongoing reclamation works with imported sand on Tekong [63], and with incineration waste as fill material on Semakau [52]. The final site—Changi—is unusual in having a high green cover while simultaneously being located on the mainland and close to the

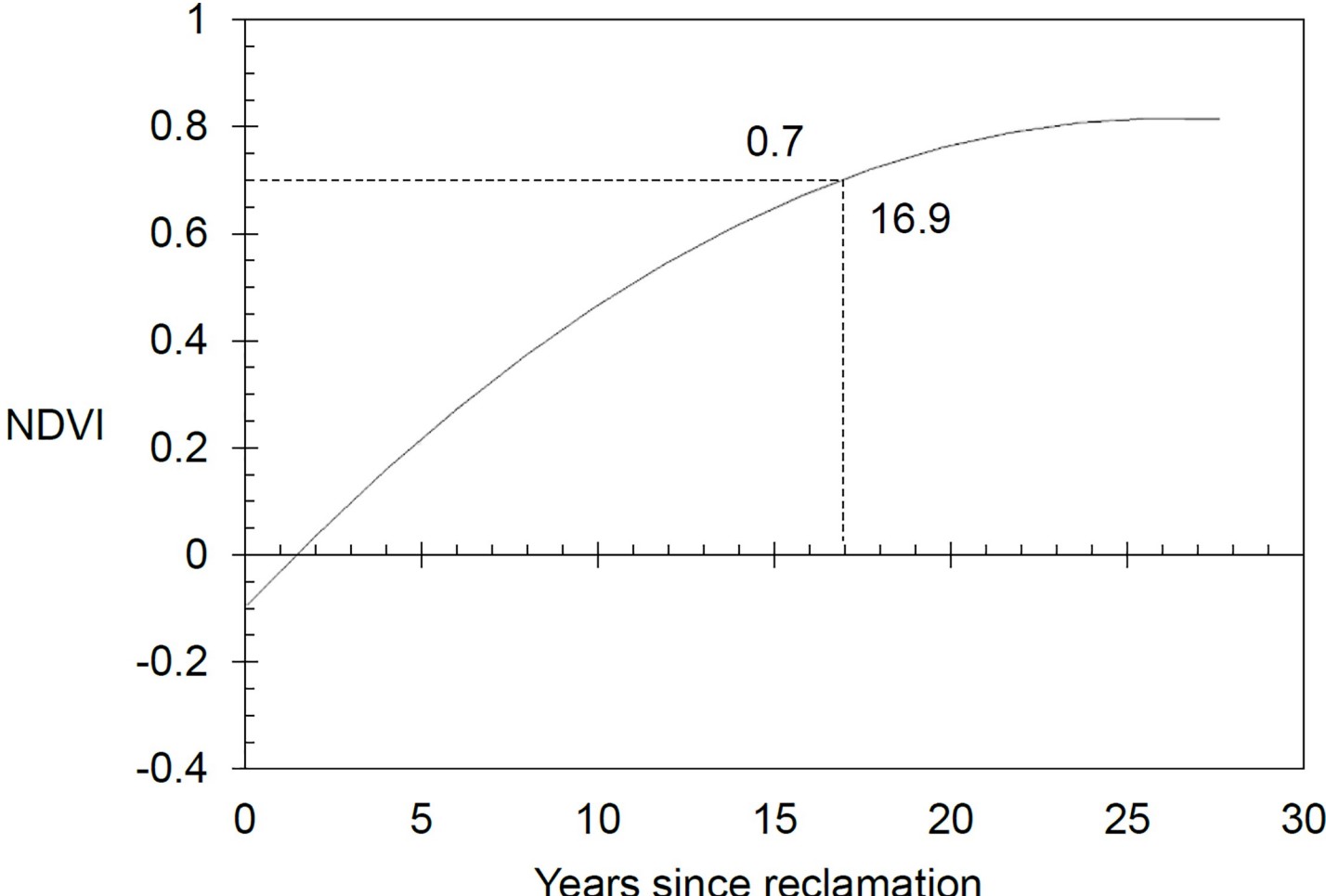

**Fig 5. The modelled trend of NDVI when land first appears.** It takes 16.9 years for NDVI to increase to 0.7 that indicates dense vegetation. Confidence intervals shown in light grey, and are very small.

**Table 3. Summary of new land reclaimed from the sea and areas with high vegetation cover.**

| Site | Mean | Median | Standard Deviation |
|---|---|---|---|
| Overall | 2001.18 | 2002 | 5.64 |
| Tuas | 2000.43 | 2001 | 6.40 |
| Jurong | 2000.52 | 2001 | 5.91 |
| Tekong | 2002.21 | 2002 | 3.56 |
| Semakau | 2000.78 | 2001 | 5.47 |
| Changi | 2003.55 | 2004 | 5.14 |

urban core. Changi is designated for a priority sector of the economy, mainly airport development, that requires only low-rise infrastructure in the form of airport runways and associated developments. However, while the airport is the priority, the development of the reclaimed region in Changi is intended to occur as part of a future development phase, with other areas available for short-term expansion [64]. The interim use of Changi as a military airbase is less spatially extensive than an international airport. Moreover, the secure and protected nature of a military airbase would mean that development and disturbance would be limited until Changi Airport Terminal 5 has been constructed and operational [65].

## Species and plant ecology on reclaimed lands

Since spontaneous vegetation that grows on reclaimed lands are pioneer species, they are, by definition, primary vegetation [66]. The trees that grow on reclaimed land creates forests that are similar in structure as Yee et al.'s [67] Waste Woodlands in Singapore, a type of secondary forest that grows on cleared lands in Singapore that is (like reclaimed lands) awaiting development. Tree species that grow are exotic such as *Acacia auriculiformis*, *Leucaena leucocephala* and *Mimosa pigra* [68]. Land reclamation over the study period occurred along the coastline of Singapore and therefore, most vegetation described here are adapted to growing along the coast that are exposed to higher winds and salinity.

Further, species occurrence depends on the fill material used. If it is subsoil that is taken from further inland, tree species that occur are similar the exotic-dominated Waste Woodlands described above. If the fill material used is marine sand, species that occur near the sea are *Canavalia cathartica* and *Ipomoea pes-caprae*, and further back where the soil is more consolidated, *Casuarina equisetifolia* and *Planchonella obovata* [69]. In the case of Changi, some planting occurred of hardy and fast-growing trees that can tolerate brackish waters. Some species used are *Casuarina equisetifolia and Terminalia catappa* [70] where their deep roots are ideal for stabilising the soil. These plantings mostly develop into monoculture plots of Casuarina Forests with an understorey shrub layer of *Lantana camara* [69]. Pulau Semakau is an exception to fill material because it is neither subsoil or marine sand but incineration waste. The landfill cells where incineration waste are dumped brings exotic weeds to the island such as *Acacia auriculiformis* that outcompetes native vegetation [71].

## Value of spontaneous vegetation in Singapore's development timeline

Spontaneous vegetation growing on reclaimed land takes about 16.9 years to develop into a lush and dense patch of vegetation (Fig 5). This duration and its implications for urban planning should be evaluated in the context of the urban development timeline of Singapore. Firstly, considering Singapore's Concept Plan 2011 [32] as an indicator of long-term land reclamation and land use planning, and secondly, the process from the start of reclamation works to achieving functional land use is around 40 years. Hence, spontaneous vegetation grown on

newly reclaimed areas in Singapore can be expected to be present for approximately 23 years with a high NDVI—a considerable length of time during which it can support biodiversity and a wide range of ecosystem services [72]. Furthermore, future land use planning is subject to change, for example in cases where downturns in the global economy have delayed construction on newly reclaimed land, giving vegetation more time to thrive [8].

Singapore has urbanised rapidly in the 20th century and continues to rejuvenate itself in the 21st century. The Concept Plan (2011) [32] outlines even more (re)development and land reclamation plans 40 to 50 years ahead. Areas with potential for reclamation include parts of Singapore's northern shores, Pulau Ubin, Pulau Tekong, Changi East, East Coast's shoreline, more of Jurong Island, and possibly the industrial islands south of Jurong Island [32]. While the current areas of spontaneous vegetation on reclaimed land may be expected to slowly decline in vegetated cover as they are being cleared for development in the coming years, new reclamation will provide new opportunities to add spontaneous ecosystems to Singapore's landscape. Future planning could harness the ecosystem service benefits provided by these new green spaces, to improve human well-being in this densely populated city.

## Benefits of leaving land to spontaneous development

Tropical vegetation are resilient ecosystems that can withstand changes caused to the natural environment by humans [73], and in this study's case, the creation of new land from the sea. For Singapore, vegetation growth on reclaimed land is a desired outcome, especially during the fallow period following reclamation. Previous work in Singapore has highlighted how vegetation that grew and ponds that accumulated on the site of the Marina Bay reclaimed land attracted migratory birds, bringing biodiversity and cultural ecosystem service benefits [8]. In addition to providing ecosystem services temporarily, the roots of plants growing on reclaimed land helps bind the soil and increase its stability [5].

Human planting projects typically have a high risk of failure, with cases of large sapling die-offs in many reforestation projects [74, 75]. For example, widespread planting of mangrove forests in The Philippines where 20,000 planted propagules died because they were not in a suitable environment [74, 76]. An alternative to planting trees is to let vegetation communities spontaneously develop. In the case of Singapore's reclamation, propagules found within the fill material or seeds brought in by the wind germinate shortly after development [77]. The species that succeed are typically hardy pioneer species such as the Casuarina tree (*Casuarina equisetifolia*) [70], which are adapted to grow in harsh environments with low soil nutrients [70]. Spontaneous vegetation not only helps land managers to save the costs and risks associated with planting, but also helps to save maintenance costs as these vegetation patches are typically not intensively pruned or weeded. Indeed, these attributes have led to spontaneous vegetation being promoted for use even in landscape architecture, more recently in public spaces of Singapore such as parks and green roofs [77, 78]. Land use managers can benefit from the estimate of spontaneous vegetation development time presented in this study. It may be possible for soil management practices or artificial propagule additions to decrease the time of vegetation development [77], giving earlier access to mature spontaneous forest vegetation.

## Vegetation monitoring with Google Earth Engine

Google Earth Engine is a powerful tool that can be used to monitor land use changes through time. There are many datasets available in the Google Earth Engine Catalog, many of which can be used to monitor land use changes from the local scale with high resolution satellites, to global studies with relatively high spatial and temporal resolution satellite systems with, for example, Sentinel satellite constellation [22]. In this study's example of Singapore, the revisit

time of Landsat is 16 days [22], but in fact, many revisits in a year are required to generate a cloud-free image of the entire country. Even so, there are tools available in Google Earth Engine to remove cloud cover to generate a cloud-free mosaic of an area of interest [79]. Landsat not only provides good spatial and temporal resolution, but also spectral resolution to calculate different vegetation indices that are useful for vegetation health monitoring [80]. A Landsat 8 image for Singapore's territorial boundary is about 100 MB in size. A Sentinel 2 image for the same areal extent, is more than ten times the storage size of Landsat 8 at 1.4 GB. This study demonstrated a simple NDVI trend analysis in the local context of Singapore on a personal computer. Google Earth Engine can also leverage on the power of cloud computing to perform Big Data analyses on geospatial imagery, which produced the Global Water Cover data inputs of this map [22, 34]. This discussion focused on pioneer species on reclaimed land in Singapore but can be extrapolated to the bioregion of the Peninsula Malaysia and larger.

## Conclusions

In conclusion, this study measured the NDVI of five reclaimed sites from the sea in Singapore to monitor pioneer vegetation establishment on it. The overall trends in NDVI were charted and modelled with a quadratic graph that shows vegetation taking about 16.9 years to reach an NDVI level of 0.7, which denotes dense and green vegetation. In Singapore, the median NDVI of the country has also increased across the 27 years of study from 0.47 in 1988 to 0.62 in 2015. This is a testament to the effectiveness of Singapore's greening measures. This study is both useful in ecology, where it estimates vegetation growth trends of pioneer species in Singapore, and geography as it analyses landscape changes to Singapore's ever-growing land area. Singapore's new reclaimed landscape from the sea are constantly in a state of flux as they are slated for development in a few years' time. Yet, before development, they serve as interesting sites for nature establishment to investigate how wild landscapes form with minimal maintenance done on them until development works begin. Even so, majority of these wild and unmanaged landscapes in Singapore are inaccessible to researchers and less so for members of the public. Satellite remote sensing is thus useful in monitoring and accessing the health and status of these unmanaged landscapes through vegetation indices like NDVI. The results of this study hopes to inform landscape managers on unmanaged vegetation growth rates and the power of Google Earth Engine for ecological landscape monitoring over time studies and of the working hypotheses.

## Supporting information

**S1 Fig. Map of Singapore's NDVI in 1988.**
(TIF)

**S2 Fig. Map of Singapore's NDVI in 2015.**
(TIF)

**S3 Fig. The trend of NDVI of each study site through the 27 years of NDVI images.**
(TIF)

## Acknowledgments

The authors would like to thank the team behind Google Earth Engine and U.S. Geological Survey for data accessibility of Landsat images.

## Author Contributions

**Conceptualization:** Leon Yan-Feng Gaw, Daniel Rex Richards.

**Data curation:** Leon Yan-Feng Gaw.

**Formal analysis:** Leon Yan-Feng Gaw, Daniel Rex Richards.

**Funding acquisition:** Daniel Rex Richards.

**Investigation:** Leon Yan-Feng Gaw, Daniel Rex Richards.

**Methodology:** Leon Yan-Feng Gaw.

**Project administration:** Daniel Rex Richards.

**Resources:** Daniel Rex Richards.

**Software:** Leon Yan-Feng Gaw.

**Supervision:** Daniel Rex Richards.

**Validation:** Leon Yan-Feng Gaw.

**Visualization:** Leon Yan-Feng Gaw.

**Writing – original draft:** Leon Yan-Feng Gaw, Daniel Rex Richards.

**Writing – review & editing:** Leon Yan-Feng Gaw, Daniel Rex Richards.

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
