## [Decision Letter · Decision Letter 0]

23 Sep 2020

PONE-D-20-26617

Development of spontaneous vegetation on reclaimed land in Singapore measured by NDVI

PLOS ONE

Dear Dr. Gaw,

Thank you for submitting your manuscript to PLOS ONE. After careful consideration, we feel that it has merit but does not fully meet PLOS ONE’s publication criteria as it currently stands. Therefore, we invite you to submit a revised version of the manuscript that addresses the points raised during the review process.

Both reviewers have indicated some revisions that are needed prior to acceptance. I have some concerns with the modelling approach that was undertaken.

We look forward to receiving your revised manuscript.

Kind regards,

Paul Pickell, Ph.D.

Academic Editor

PLOS ONE

Journal Requirements:

3. We note that Figures in your submission contain map/satellite images which may be copyrighted.

a. You may seek permission from the original copyright holder of the Figures to publish the content specifically under the CC BY 4.0 license. 

4. Please ensure that you include a title page within your main document. We do appreciate that you have a title page document uploaded as a separate file, however, as per our author guidelines (http://journals.plos.org/plosone/s/submission-guidelines#loc-title-page) we do require this to be part of the manuscript file itself and not uploaded separately.

Additional Editor Comments:

There is no indication that the authors considered the confounding effect of temporal autocorrelation in their models. In the authors’ conceptual model of land reclamation, NDVI is temporally dependent. This leads me to believe that the R-squared values of the models are overestimated and the p-values are difficult to interpret because the assumption of data independence has been violated. Please explain the temporal autocorrelation with an autocorrelogram and adjust your modelling techniques based on the autocorrelation structure of the data. Alternatively, the authors can try non-parametric models such as the Theil-Sen estimator and the Mann-Kendall rank sum test, which have both been extensively used and tested in the literature with NDVI time series.

Reviewers' comments:

Reviewer's Responses to Questions

**Comments to the Author**

1. Is the manuscript technically sound, and do the data support the conclusions?

Reviewer #1: Yes

Reviewer #2: Yes

2. Has the statistical analysis been performed appropriately and rigorously? 

Reviewer #1: Yes

Reviewer #2: Yes

3. Have the authors made all data underlying the findings in their manuscript fully available?

Reviewer #1: Yes

Reviewer #2: Yes

4. Is the manuscript presented in an intelligible fashion and written in standard English?

Reviewer #1: Yes

Reviewer #2: Yes

5. Review Comments to the Author

Reviewer #1: First of all, I think this is a very important paper and definitely should be published. The analysis and identification of natural growth areas will become increasingly important in the future and this paper further improves the knowledge needed to deal with these questions.

The arbitrary 0.7 threshold is ok and follows the values that normally are used for tropical areas. And the 15+ (16.8) years that have been found also follows the major scientific references like the ones studied by Robin Chazdon in her book "Second Growth: The Promise of Tropical Forest Regeneration in an Age of Deforestation" and by other important papers.

Despite that, I have an issue with the results. The paper says that the NDVI in reclaimed areas has increased at a more rapid rate than the national median. But, the national median calculation also considers the reclaimed areas or not? Depending on the answer, I think it will be better to show the national median without those areas to have a real perception of the growths.

Reviewer #2: The manuscript does present a good overall view of the current status of Singapore's vegetation distribution in specific areas. Using freely accessible datasets is also advantageous, keeps research open and reproducible. Methods used in the manuscript are straightforward and using the NDVI as a proxy to measure vegetation keeps matters simple. Even though the paper is not using novel methods, it highlights important phenomena that affect the country, which is highly urbanised and built up, furthermore resulting in situations that are unique to Singapore and require targeted action (such as dealing with reclaimed land).

The paper seems to use terms "Land use" and " land cover" interchangeably. The distinction needs to be made clear. In the Singaporean context, land use seems to make more sense, as most of the area is artificially built up and then it simply begs the question what exactly is the land used for - for example, airport or housing. However, the background information section uses land cover only.

The authors mention using Google Earth Engine, however, the script and code used was not attached. It makes sense to attach it in the supplementary material.

The authors also mention development of plant species on reclaimed land in the abstract and other parts in the paper. It would be meaningful to mention the exact type of vegetation growing on these reclaimed areas. The paper does highlight the ecological importance of the work, however, background information regarding the ecological state of the study area is somewhat lacking. Even though the aim of the paper is not to quantify vegetation, more background about the type of plant species and implications for ecology should be highlighted more. Also

English requires some revision, there is a lot of switching between active and passive voice throughout.

6. PLOS authors have the option to publish the peer review history of their article (what does this mean?). If published, this will include your full peer review and any attached files.

Reviewer #1: **Yes: **Eduardo Lacerda

Reviewer #2: No

---

## [Author Response · Author response to Decision Letter 0]

4 Dec 2020

Reviewer 1

First of all, I think this is a very important paper and definitely should be published. The analysis and identification of natural growth areas will become increasingly important in the future and this paper further improves the knowledge needed to deal with these questions.

Response: We thank the Reviewer for the positive comments and encouragement. We hope that our findings will highlight the importance of spontaneous vegetation growth on reclaimed lands, especially in expanding coastal cities.

The arbitrary 0.7 threshold is ok and follows the values that normally are used for tropical areas. And the 15+ (16.8) years that have been found also follows the major scientific references like the ones studied by Robin Chazdon in her book "Second Growth: The Promise of Tropical Forest Regeneration in an Age of Deforestation" and by other important papers.

Response: We have added Chadzon’s work as a reference and cited it in our manuscript line 177.

Despite that, I have an issue with the results. The paper says that the NDVI in reclaimed areas has increased at a more rapid rate than the national median. But, the national median calculation also considers the reclaimed areas or not? Depending on the answer, I think it will be better to show the national median without those areas to have a real perception of the growths.

Response: This suggestion is valid and we have included a trend line that shows the national median of NDVI that excludes newly reclaimed areas. Therefore, Figure 3 will have three NDVI trend lines of (1) the combined reclaimed and other terrestrial areas, (2) reclaimed areas only, (3) non-reclaimed terrestrial areas only. The slope for the total national and total national excluding reclaimed areas are similar.

Reviewer 2

The manuscript does present a good overall view of the current status of Singapore's vegetation distribution

in specific areas. Using freely accessible datasets is also advantageous, keeps research open and reproducible. Methods used in the manuscript are straightforward and using the NDVI as a proxy to measure vegetation keeps matters simple.

Response: We thank the Reviewer for the positive comments on our manuscript. We especially want to highlight the technological advances that have been made in remote sensing and how accessible they are now to researchers.

Even though the paper is not using novel methods, it highlights important phenomena that affect the country, which is highly urbanised and built up, furthermore resulting in situations that are unique to Singapore and require targeted action (such as dealing with reclaimed land). The paper seems to use terms "Land use" and " land cover" interchangeably. The distinction needs to be made clear. In the Singaporean context, land use seems to make more sense, as most of the area is artificially built up and then it simply

begs the question what exactly is the land used for - for example, airport or housing. However, the background information section uses land cover only.

Response: We agree with the Reviewer that the appropriate term in the urban context of Singapore is ‘land use’ and have standardised our manuscript to use only that term.

The authors mention using Google Earth Engine, however, the script and code used was not attached. It makes sense to attach it in the supplementary material.

Response: We will make our data and code available on a Figshare, an online repository of scientific work, once the manuscript is accepted for publication. This is in line with PLOS One’s open access policy.

The authors also mention development of plant species on reclaimed land in the abstract and other parts in the paper. It would be meaningful to mention the exact type of vegetation growing on these reclaimed areas. The paper does highlight the ecological importance of the work, however, background information regarding the ecological state of the study area is somewhat lacking. Even though the aim of the paper is not to quantify vegetation, more background about the type of plant species and implications for ecology should be highlighted more.

Response: We have added a discussion section to elaborate on the plant ecology and its state in reclaimed lands in Singapore (lines 287–307).

Also English requires some revision, there is a lot of switching between active and passive voice throughout.

Response: We have read through the manuscript again and have standardised our writing to a passive voice writing style.

---

## [Decision Letter · Decision Letter 1]

26 Dec 2020

Development of spontaneous vegetation on reclaimed land in Singapore measured by NDVI

PONE-D-20-26617R1

Dear Dr. Richards,

We’re pleased to inform you that your manuscript has been judged scientifically suitable for publication and will be formally accepted for publication once it meets all outstanding technical requirements.

Kind regards,

Paul Pickell, Ph.D.

Academic Editor

PLOS ONE

Additional Editor Comments (optional):

Reviewers' comments:

Reviewer's Responses to Questions

**Comments to the Author**

1. If the authors have adequately addressed your comments raised in a previous round of review and you feel that this manuscript is now acceptable for publication, you may indicate that here to bypass the “Comments to the Author” section, enter your conflict of interest statement in the “Confidential to Editor” section, and submit your "Accept" recommendation.

Reviewer #1: All comments have been addressed

Reviewer #2: All comments have been addressed

2. Is the manuscript technically sound, and do the data support the conclusions?

Reviewer #1: Yes

Reviewer #2: Yes

3. Has the statistical analysis been performed appropriately and rigorously? 

Reviewer #1: Yes

Reviewer #2: N/A

4. Have the authors made all data underlying the findings in their manuscript fully available?

Reviewer #1: Yes

Reviewer #2: Yes

5. Is the manuscript presented in an intelligible fashion and written in standard English?

Reviewer #1: Yes

Reviewer #2: Yes

6. Review Comments to the Author

Reviewer #1: The revisions contributed to a significant improvement in the results. Even with similar slopes, it's good to have extra information about the differences from the national median with and without the reclaimed areas. And also the updated national median NDVI value.

Reviewer #2: The paper satisfied all the comments and concerns. Good job on keeping all your findings and methods openly accessible and available.

7. PLOS authors have the option to publish the peer review history of their article (what does this mean?). If published, this will include your full peer review and any attached files.

Reviewer #1: No

Reviewer #2: No

---

## [Editor Report · Acceptance letter]

14 Jan 2021

PONE-D-20-26617R1 

Development of spontaneous vegetation on reclaimed land in Singapore measured by NDVI 

Dear Dr. Gaw:

I'm pleased to inform you that your manuscript has been deemed suitable for publication in PLOS ONE. Congratulations! Your manuscript is now with our production department. 

Kind regards, 

on behalf of

Dr. Paul Pickell 

Academic Editor

PLOS ONE